# Does ischemic preconditioning really improve performance or it is just a placebo effect?

**Hiago L. R. de Souza**[1], **Rhaí A. Arriel**[1], **Gustavo R. Mota**[2], **Rodrigo Hohl**[1], **Moacir Marocolo**[1] *

**1** Physiology and Human Performance Research Group, Department of Physiology, Federal University of Juiz de Fora, Juiz de Fora, Minas Gerais, Brazil, **2** Exercise Science, Health and Human Performance Research Group, Department of Sport Sciences, Institute of Health Sciences, Federal University of Triangulo Mineiro, Uberaba, Minas Gerais, Brazil

* isamjf@gmail.com

## Abstract

This study examined the effects of a simultaneous ischemic preconditioning (IPC) and SHAM intervention to reduce the placebo effect due to a priori expectation on the performance of knee extension resistance exercise. Nine moderately trained men were tested in three different occasions. Following the baseline tests, subjects performed a first set of leg extension tests after the IPC (3 X 5 min 50 mmHg above systolic blood pressure) on right thigh and the SHAM (same as IPC, but 20 mmHg) on left thigh. After 48 hours, the subjects performed another set of tests with the opposite applications. Number of repetitions, maximal voluntary isometric contraction (MVIC) and perceptual indicators were analyzed. After IPC and SHAM intervention performed at the same time, similar results were observed for the number of repetitions, with no significant differences between conditions (baseline x IPC x SHAM) for either left (p = 0.274) or right thigh (p = 0.242). The fatigue index and volume load did not show significant effect size after IPC and SHAM maneuvers. In contrast, significant reduction on left tight MVIC was observed (p = 0.001) in SHAM and IPC compared to baseline, but not for right thigh (p = 0.106). Results from the current study may indicate that applying IPC prior to a set of leg extension does not result in ergogenic effects. The placebo effect seems to be related to this technique and its dissociation seems unlikely, therefore including a SHAM or placebo group in IPC studies is strongly recommended.

## Introduction

Ischemic preconditioning (IPC) is a noninvasive intermittent local ischemia performed on the subject's limbs for a brief period of time [1]. IPC effectiveness in exercise performance has been demonstrated in studies carried out with low to moderate fitness level subjects under laboratory conditions [2]. However, recent studies presented heterogeneous responses after IPC probably due to the variability of protocols, and several aspects of the technic such as cuff pressures, number of cycles, occlusion and reperfusion times, the time between the IPC and the beginning of the exercise [3, 4].

**Funding:** This research was funded by the Federal University of Juiz de Fora - UFJF, supporting this

study and APC. The funders had no role in study design, data collection and analysis, decision to publish, or preparation of the manuscript.

**Competing interests:** The authors have declared that no competing interests exist.

Despite IPC having been extensively tested due its low cost and easy application, mechanisms for these effects on performance are still lacking. There are arguments and evidences questioning the peripheral/local effect of IPC [5, 6] pointing a potential placebo effect [2]. In this sense, recent research have demonstrated no IPC effect on peripheral fatigue and neuromuscular function compared to sham IPC (SHAM) after submaximal isometric [5] or all out isokinetic tests [6]. Therefore, minimizing potential confounding factors like placebo effect, warrant further investigations.

Placebos are normally considered a physiologically inert treatment [7], however, the placebo effect in sports and exercise can be also defined as a neurobiological-cognitive response that enhances performance as a consequence of SHAM treatment, involving inert drugs, equipment or even verbal encouragement [8]. This phenomenon has been experienced by athletes of different levels [9, 10].

The subject's beliefs about a beneficial or harmful treatment can significantly modulate performance [11] and may be explained by a psychophysiological top-down model that associates the declarative or nondeclarative mental processing at the level of the cerebral cortex with somatic and behavioral responses [12]. Recent results seems to point to this way, where IPC and SHAM interventions showed similar effects on resistance exercise (RE) performance, evidencing a possible top-down effect due to the cognitive appraisal of the context of cuff maneuvers [13]. Moreover, these results related to the placebo effect seems to be aligned with Pollo, Carlino [14] that observed a top-down modulation on the performance of leg extension increasing muscle work after an ergogenic placebo treatment.

Taken together, the IPC intervention by itself could increase the expectation of improved performance linked to the sensation of pressure applied to the limb plus the instruction provided by the experimenter, consequently inducing a placebo effect. Thus, carrying out a simultaneous application of IPC and SHAM could be a good alternative that may minimize the potential placebo effects in this intervention, promoting divergent expectations in the subjects and presenting new insights on the real effect of the IPC.

Therefore, based on assumption that IPC may be dependent of a priori conditioning or expectancy to induce a placebo effect, the aim of this study was to compare the effects of a simultaneous application of IPC and SHAM on the RE performance. Based on recent literature [4, 13], it was hypothesized that both IPC and SHAM cause similar outcomes in RE performance, translating into a placebo effect.

## Materials and methods

### Subjects

Based on a recent systematic review [4], we have selected studies which investigated the effect of IPC on RE and calculated the effect sizes obtaining the mean of 0.54. Then, it was estimated a sample size of at least 8 participants based on the effect size, $\alpha$ level of 0.05, power (1-$\beta$) of 0.80, two interventions and three repeated measurements, (G*Power 3.1.9.2, Heinrich-Heine-Universität Düsseldorf, Düsseldorf, Germany; http://www.gpower.hhu.de/). To guarantee a statistical power we have included nine healthy and young men (22.4 ± 3.4 yrs; 176.7 ± 5.5 cm; 74.7 ± 5.6 kg; 11.8 ± 4.2 body fat %; 80.5 ± 10.4 1RM left thigh [kg]; 82.7 ± 11.2 1RM right thigh [kg]; 88.8% right-handed). The participants declared 3.4 ± 1.7 years of experience in RE and were involved in 4.0 ± 0.9 h wk$^{-1}$ of regular training. Exclusion criteria included the following: (a) smoking history during the previous 3 months, (b) presence of any cardiovascular or metabolic disease, (c) systemic hypertension (≥140/90 mmHg or use of antihypertensive medication), (d) use of creatine supplementation, (e) use of anabolic steroids, drugs or medication with potential impact in physical performance (self-reported), or (f) recent

musculoskeletal injury. Experimental procedures were approved by the Human Research Ethics Committee of the Federal University of Juiz de Fora Human Research Ethics Committee, approval number 33413214.1.0000.5154 and were performed according to the Declaration of Helsinki. Also, all procedures were explained to subjects, and they signed an informed consent form before data collection.

## Design

This study carried out a within-person RE performance comparison of left and right limbs after simultaneous IPC and SHAM application on both thighs. This study design (Fig 1) allowed each subject to serve as his own control. At baseline, subjects performed a unilateral maximal voluntary isometric contractions test (MVIC) and after a rest interval of 3 minutes performed 3 sets with 2-minute rest of unilateral leg extension. Following the baseline tests, subjects performed identical baseline RE tests protocol after the IPC on right thigh and the SHAM on left thigh (randomized). Following 48 hours (avoiding the possible effect of a second window of IPC), the subjects performed another RE tests protocol with the opposite IPC and SHAM applications.

 The data collection was individualized, with no contact between study subjects. All tests were conducted by the same experienced researcher, at the same period of the day (16:00h-18:00h). The subjects were instructed to refrain from coffee and alcohol consumption 24 hours before and to practice strenuous exercises 48 hours preceding testing. Subjects were familiarized with all tests before we collect the data to decrease the effect of learning.

## Procedures

**IPC and SHAM intervention.** Occlusion cuffs (96 x 13 cm; Komprimeter Riester®, Jungingen, Germany) were placed at the upper thigh of the subject. In the IPC protocol, the cuff was inflated to 50 mmHg above systolic blood pressure of each subject for 5 min and repeated three times (i. e., 3 x 5 min occlusion/5 min reperfusion), as previously shown being a sufficient pressure to obstruct blood flow [15]. In the SHAM protocol, subjects received an identical intervention, but the cuff was inflated to only 20 mmHg. Both IPC/SHAM were performed randomized and interspersed on the subject thigh (i.e., the cuffs were applied simultaneously, but when one thigh received their respective pressure the other was under reperfusion), resulting in a total intervention time of 30 min. To equalize cognitive appraisal of intervention and avoid a nocebo effect [16], subjects were informed about the testing of two external pressure conditions and that both could improve performance. The effectiveness of occlusion/non-

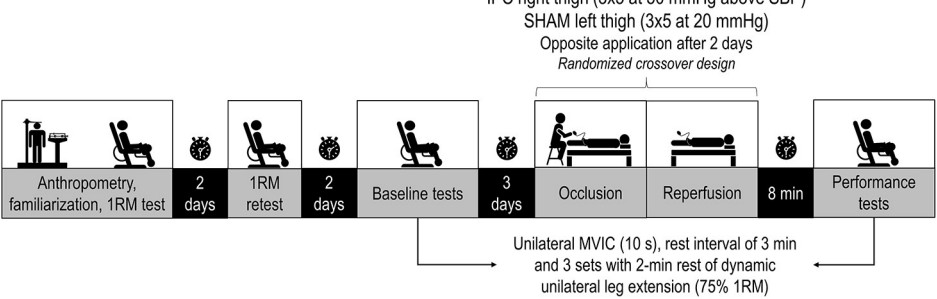

**Fig 1. Experimental design of the study.** MVIC, maximal voluntary isometric contractions; IPC, ischemic preconditioning; SHAM, cuff application with low pressure; SBP, systolic blood pressure; 1RM, one repetition maximum; mmHg, millimeter of mercury.

occlusion was checked continuously by auscultation of the tibial anterior artery [16]. During IPC and SHAM application, the principal researcher left the room to remain blinded to interventions and to ensure unbiased data collection and analysis.

**Unilateral 1 repetition maximum test and retest.** The leg extension tests were performed on a leg extension machine (Model Element; Technogym, Gambetolla, Italy). First, the subjects performed a general warm-up (3–5 minutes of light activity, i.e., walking, articular movements without loads, and mild static stretching) involving the tested muscle group [17], followed by a specific warm-up (1 set of 10–12 repetitions of unilateral leg extension with 30% of body mass load) with a movement cadence of 1 second for concentric and 2 seconds for eccentric phase. Rest intervals of 5 minutes were established between each attempt to 1RM. No more than 5 attempts were necessary to find 1RM load. The 1RM was established when the subject was able to perform one completed repetition of the movement (concentric and eccentric phase in the predetermined cadence) but was unable to perform a second repetition without assistance. The 1RM test procedure was performed according to National Strength and Conditioning Association recommendation [18]. A metronome (DM50; Seiko, Tokyo, Japan) was used to ensure the correct cadence of the movement. The 1RM retest was performed on a separate day, 48 hours after the first trial. The intraclass correlation coefficient was calculated for 1RM test and retest (left thigh: 0.982, confidence interval = 0.922–0.996; right thigh: 0.996, confidence interval = 0.971–0.999).

**Maximal voluntary isometric contractions test and repetition sets until concentric failure.** The 8-minute break between the IPC/SHAM application (Fig 1) and RE sets involved removal of the cuffs and conducting a specific warm-up of 1 set of 10–15 repetitions of the leg extension with 50% of the 1RM load. MVIC was performed using a load cell (SDS1000, Miotool system) according previously described methods [19]. Briefly, 3 attempts of 10 seconds with 1-minute rest were performed with hip and knee angles fixed at 100° and 90°, respectively. An ankle strap was placed 2 cm proximal to the medial malleolus, and a load cell was positioned perpendicular to tibiae alignment. Participants were instructed to contract as hard as possible and the mean peak of the 3 attempts was considered for data analysis. Subsequently, after a rest interval of 3 minutes, the subjects performed 3 sets with 2-minute rest of unilateral leg extension until momentary failure with the predetermined 75% 1RM load. Momentary failure was considered when the subjects reached the end point where despite attempting to do, so they could not complete the concentric portion of their current repetition without deviation from the prescribed form of the exercise [20]. Movement cadence in all sets was controlled (1 second for concentric and 2 seconds for eccentric phase) by a metronome (DM50; Seiko, Tokyo, Japan). The fatigue index (FI) was considered as the degree of decrease of number of repetitions between the first and third sets of leg extension [21], expressed as percentage: FI = (set1-set3)/set-1 x 100. Volume load was calculated as (sets x repetitions x load).

**Perception of the treatment and expectancy.** On the two days of treatment and tests, expectancy about the effect of previous IPC/SHAM interventions on RE performance was registered. The subjects answered the following question: "Did you expect any effect on your performance? (no influence/positive influence/negative influence)" [22].

**Perceptual measurements.** Before all performance tests, the subjects answered the perceived recovery status scale (PRS) [23] indicating a score ranging from 0 to 10 to make sure that the recovery level was the same.

The pain perception was assessed through a numerical rating scale immediately after the simultaneous IPC/SHAM applications. The numerical rating scale consists of a 0–10 score scale [24], where the lowest value means "no pain" and the highest value means "unbearable pain." The following instructions were presented after cuff removal: "Select a single number that best represents the pain intensity felt during this intervention".

The perceived exertion was assessed through the Omnibus-Resistance Exercise Scale [25] (values from 0 to 10) at the end of each set.

**Statistical analysis.** The Shapiro-Wilk test was applied to verify the normality of data distribution. The nonparametric ANOVA (Friedman test) followed by a post hoc Dunn's test was conducted for comparison for the number of repetitions and perceived exertion. A repeated-measures one-way ANOVA followed by Bonferroni post hoc or Friedman test followed by Dunn's post hoc, when necessary, was conducted for analysis of MVIC, volume load and FI. The Wilcoxon's signed-rank test was conducted for comparison of pain between protocols. Effect size (ES) was calculated with magnitude classified as trivial ($<0.35$), small ($0.35–0.80$), moderate ($0.80–1.50$) and large ($>1.5$) based on guidelines specifically for resistance trained individuals [26]. The significance level was 0.05, and the software used for data analysis was GraphPad (Prism 8.0.0; San Diego, CA, USA). For parametric tests the data are expressed as mean and standard deviation and for nonparametric tests as median and lower-upper 95% CI of median.

## Results

Comparisons were made between left or right thigh x baseline tests.

Similar results between both thighs were observed for the number of repetitions (Fig 2A), with no significant differences between conditions for either left [$\chi^2(2) = 2.818$, $p = 0.274$] or right thigh [$\chi^2(2) = 3.500$, $p = 0.242$]. In contrast, significant reduction between conditions were observed for left thigh on MVIC [$F(1.752,14.02) = 14.05$, $p = 0.001$], but not for right thigh [$\chi^2(2) = 4.667$, $p = 0.106$] (Fig 2B).

The fatigue index did not show significant effect between conditions for either left [$F(1.749,13.99) = 2.668$, $p = 0.109$] or right thigh [$F(1.419,11.35) = 0.025$, $p = 0.937$]. The same was observed for volume load, that did not show significant effect for either left [$F(1.661,13.29) = 1.266$, $p = 0.306$] or right thigh [$F(1.332,10.65) = 1.424$, $p = 0.270$].

The rating of perceived exertion scores did not show significant differences [Baseline = 8.5 (7.5–9.0); First day = 8.0 (7.0–9.0); Second day = 8.0 (7.0–9.0); $p = 0.146$].

Just before the bouts, the perceived recovery scores did not show significant differences [Baseline = 8.0 (8.0–9.0); First day = 8.0 (6.0–9.0); Second day = 9.0 (7.0–9.0); $p = 0.442$].

Conversely, the ratings of perceptions of pain showed significant differences between conditions for left [IPC: 6.0 (4.0–7.0); SHAM: 1.0 (0.0–2.0); $p = 0.003$] and right thigh [IPC: 6.0 (3.0–8.0); SHAM: 1.0 (0.0–3.0); $p = 0.007$].

The magnitude of treatment effects according to baseline measures are presented in Table 1.

The expectancy on their performance after receiving the intervention (simultaneous IPC/SHAM cuff conditions) presented heterogeneous results. On the first day of tests four subjects (44.5%) expected no influence, four subjects (44.5%) expected positive influence and one subject (11.0%) expected negative influence for IPC/SHAM. On the second day, three subjects (33.5%) expected no influence, four subjects (44.5%) expected positive influence and two subject (22.0%) expected negative influence. All participants stated that they noticed a difference between the two pressure cuff conditions.

## Discussion

This study is the first to investigate the simultaneous application of IPC and SHAM in RE performance. It was hypothesized that both IPC and SHAM would show similar outcomes in RE performance. Our main finding is that IPC or SHAM did not improve performance of any muscular variables of knee extension exercise on both thighs, suggesting that improvements in

**A**

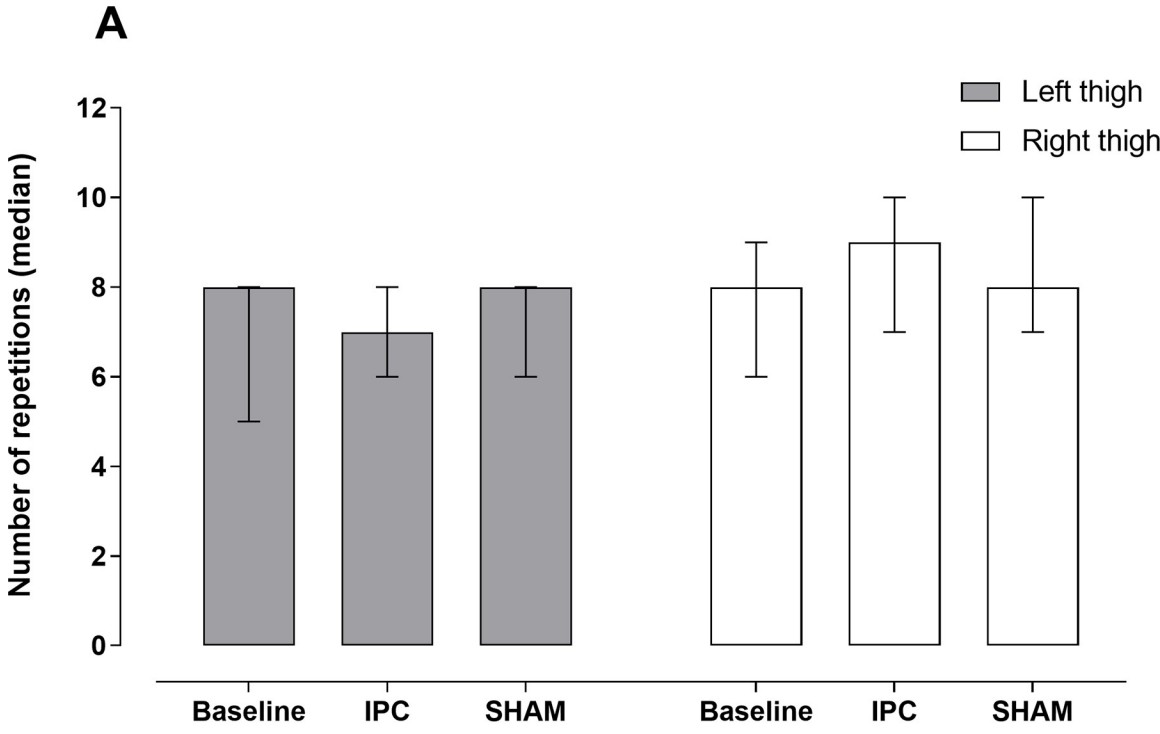

**B**

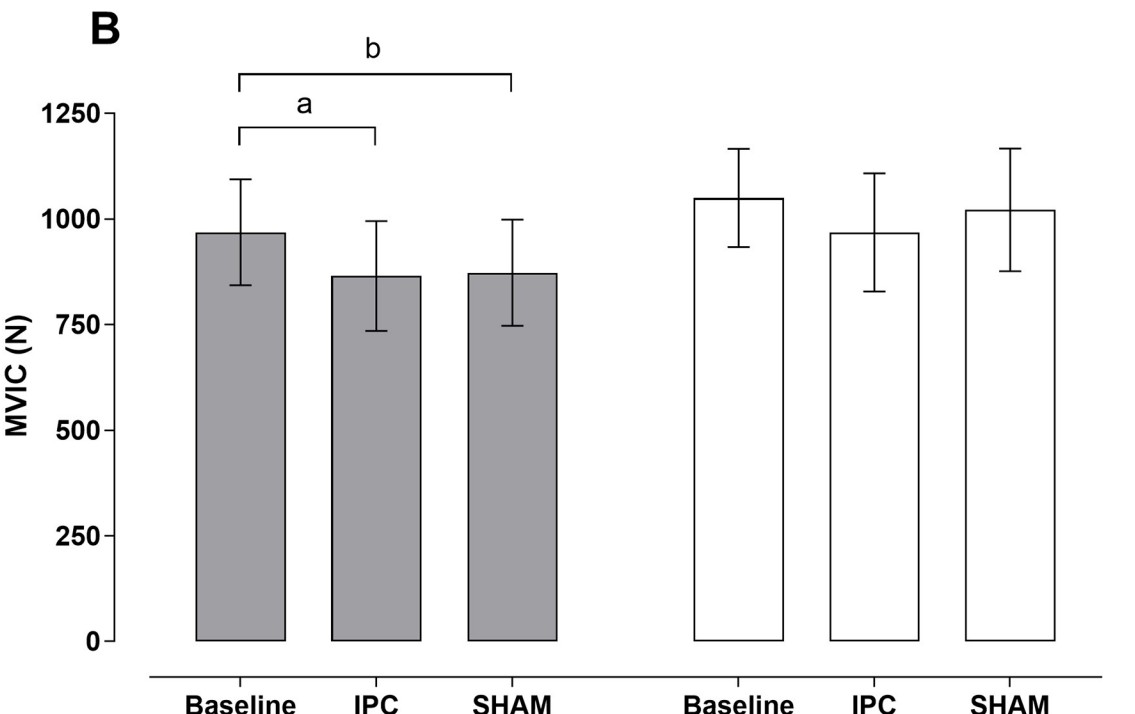

**Fig 2. A) number of repetitions between conditions baseline, IPC and SHAM for the left and right thigh. B) Maximal voluntary isometric contraction between conditions baseline, IPC and SHAM for the left and right thigh**. [a]p = 0.010, ES: -0.83; [b]p = 0.004, ES: -0.76. IPC, ischemic preconditioning; SHAM, cuff application with low pressure.

**Table 1. Effect size according to baseline measures.**

| Variable | Treatment | Effect size | |
|---|---|---|---|
| | | Left thigh | Right thigh |
| Number of repetitions | IPC | -0.16 | 0.33 |
| | SHAM | 0.16 | 0.24 |
| MVIC | IPC | -0.83 | -0.70 |
| | SHAM | -0.76 | -0.24 |
| Fatigue Index | IPC | -0.54 | -0.05 |
| | SHAM | -0.85 | -0.03 |
| Total load | IPC | 0.01 | 0.40 |
| | SHAM | 0.17 | 0.31 |

MVIC, maximal voluntary isometric contraction; IPC, ischemic preconditioning; SHAM, cuff application with low pressure.

exercise performance after IPC may be mediated by a central placebo effect instead of a local ergogenic effect. We posit that the simultaneous application of both interventions may have nullified the possible placebo effects. Moreover, we observed that the SHAM and IPC MVIC of the left thigh was lower than baseline (Fig 2B). This outcome supports the statement that IPC does not benefit exercise modes lasting up to 10 s [1]. In relation to the dominance limb, we minimize this possible effect due to randomization during the application of the intervention.

We also observed no significant effects on number of repetitions and volume load. Extending these results, Valenzuela, Martin-Candilejo [27] also did not observed improvements neither in number of repetitions nor in load-velocity relationship with 30, 50 or 70% of 1RM during bench press exercise. Additionally, Halley, Marshall [6], performed an application of IPC (3x5 min of ischemia/reperfusion) and verified its influence on voluntary activation, maximal torque production, electromyography and muscle oxygenation following a 3-min all out maximal power dynamic single leg extension test. The authors did not observe effect of IPC on any of the variables analyzed. Taken together, these recent evidences may suggest that IPC acutely applied has negligible local ergogenic effects on RE training volume variables, peripheral fatigue or neuromuscular function.

Regarding perceptual measurements, no differences between IPC and SHAM were observed. The IPC appears to have minimal influence on subjective markers of physiological stress, such perceived exertion. This result does not seem to be restricted to RE [13, 17], but it was also reported in different exercise-modes [22, 28]. Additionally, the PRS scale without significant differences suggest a control of possible bias related to the recovery of the evaluated subjects.

The possible mechanisms underlying the IPC to improve exercise performance are largely speculative, with insufficient data demonstrating clear physiological effects. There are assumptions that IPC could act locally on the tissue that received ischemia or acting remotely on other organs [29]. In this context, neural pathways are speculated for an end effect through afferent signaling feedback to the brain and then to the target organ (e. g. heart) or by arousing humoral pathways through agents in bloodstream and thus possibly improving $O_2$ delivery via vasodilatory mechanisms [3, 29]. Nevertheless, statements that IPC could mediate physiologic efficiency such as recruitment of additional motor units, enhance muscle deoxygenation and improve skeletal muscle blood flow [1] seems to be refuted based on recent research that have not observed such improvements [5, 6]. On the other hand, the training status may play a role in the IPC response [4]. Only a few studies have investigated the effect of IPC on highly trained

individuals (e.g., [30, 31]). The majority part of the studies tested less fit subjects, which makes them more susceptible to respond to the IPC intervention and, at least in part, can justify the positive results previously found in the literature.

It is noteworthy that most explanations for IPC effects are often transposed from a clinical setting, in a non-exercise model, as an attempt to justify increases in performance without clear physiological evidences [4, 29]. Because a placebo component is strongly linked to IPC [2, 4], a more holistic and integrative framework should be taken into account. It is noted that higher magnitude of placebo effect occurs more frequently in subjects with lower training status compared to elite athletes, which are more adapted to intervention, manipulation or other strategies to increase performance. Furthermore, the placebo effect is based on what the subject believes and its effectiveness is dependent on prior conditioning and or expectation [8]. This statement lead to a bidirectional interaction between professional (i.e. researcher or coach) and receiver in the ability to generate a placebo effect [8, 11], since a relationship of trust between the two sides are essential [11]. Regarding IPC, a modification in performance could be considerably associated with the conditioning and expectation of the intervention, which makes sense when is supported by a scarcity of physiological effects [4, 29]. It remains to know if the acute effects provided by the IPC/SHAM previously reported, presumably a placebo effect, would be observed again with its continuous use. We do not yet have adequate evidence for this, although a previous study points to an effect that fade over time with repeated applications [17]. In fact, this scenario corroborates a previous statement that not all subjects are responsive to placebo, or at least, not all the time [11].

In addition, according to the expectation theory related to the placebo effect, receive an information about the intervention prior to its manipulation, in this case IPC, could induce a response to this intervention based on what the subject is led to think that will happen [32]. In this sense, by equalizing the cognitive appraisal of the intervention informing the subjects that the applied external pressure increases performance, the researcher/coach could induce a placebo response. This could be the case of the most of the IPC experimental protocols.

An undeniable limitation of IPC research is the difficult to blind the subjects due to great differences between the IPC and SHAM pressures. In this regard, the subjects were informed about the testing of two external pressure conditions and that both could equally improve performance. As expected, all subjects have noticed the difference between IPC and SHAM but this perception did not influence the RE performance when both IPC and SHAM were applied simultaneously. In this sense, any possible placebo effect of IPC and SHAM were randomly distributed in this experimental design as suggested by the 'expectancy of performance' after receiving the intervention, however, the IPC maneuver did not produce any local systematic effect on RE performance. Additionally, since our protocol is limited to the leg extension exercise, we cannot extrapolate our findings to others type of exercise.

Last but not least, we highlight some methodological issues to allow advances in IPC and exercise studies. Based on a recent review [4], only 17.77% of IPC studies (8 out of 45) used a control/baseline, placebo and IPC experimental design. Future studies should consider carrying out experiments with this 3-conditions design, since we must assume the placebo condition as an active treatment and not a passive control [9, 33]. Likewise, to evaluate the subject perceptions and beliefs about effectiveness of a possible ergogenic agent [33], could provide new insights on what was previously provocatively attributed to IPC believers and nonbelievers [34].

## Conclusion

The IPC administration prior knee extension resistance exercise has no local effect and did not potentiate either the number of repetitions or MVIC in moderately RE trained men. A

motivational effect seems to be related to this technique and, researchers and coaches should take this into account, before using IPC as an exercise enhancement tool, once a dissociation of IPC and placebo effect seems unlikely. Also, the inclusion of SHAM group in IPC studies is strongly recommended.

## Supporting information

**S1 Data.**
(XLSX)

## Author Contributions

**Conceptualization:** Hiago L. R. de Souza, Rodrigo Hohl, Moacir Marocolo.

**Data curation:** Hiago L. R. de Souza, Rhaí A. Arriel.

**Formal analysis:** Hiago L. R. de Souza, Rhaí A. Arriel.

**Investigation:** Hiago L. R. de Souza, Rhaí A. Arriel.

**Methodology:** Hiago L. R. de Souza, Gustavo R. Mota, Rodrigo Hohl, Moacir Marocolo.

**Project administration:** Gustavo R. Mota, Rodrigo Hohl, Moacir Marocolo.

**Resources:** Hiago L. R. de Souza.

**Supervision:** Moacir Marocolo.

**Validation:** Moacir Marocolo.

**Visualization:** Moacir Marocolo.

**Writing – original draft:** Hiago L. R. de Souza, Rhaí A. Arriel, Rodrigo Hohl, Moacir Marocolo.

**Writing – review & editing:** Hiago L. R. de Souza, Rhaí A. Arriel, Gustavo R. Mota, Rodrigo Hohl, Moacir Marocolo.

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
