## [Decision Letter · Decision Letter 0]

29 Mar 2021

PONE-D-21-05373

Does ischemic preconditioning really improve performance or it is just a placebo effect?

PLOS ONE

Dear Dr. Marocolo,

Thank you for submitting your manuscript to PLOS ONE. After careful consideration, we feel that it has merit but does not fully meet PLOS ONE’s publication criteria as it currently stands. Therefore, we invite you to submit a revised version of the manuscript that addresses the points raised during the review process.

We look forward to receiving your revised manuscript.

Kind regards,

Daniel Boullosa

Academic Editor

PLOS ONE

Journal Requirements:

2. In the Methods section, please provide additional details regarding how participants were recruited for the study.

Reviewers' comments:

Reviewer's Responses to Questions

**Comments to the Author**

1. Is the manuscript technically sound, and do the data support the conclusions?

Reviewer #1: Yes

Reviewer #2: Yes

2. Has the statistical analysis been performed appropriately and rigorously? 

Reviewer #1: Yes

Reviewer #2: Yes

3. Have the authors made all data underlying the findings in their manuscript fully available?

Reviewer #1: Yes

Reviewer #2: Yes

4. Is the manuscript presented in an intelligible fashion and written in standard English?

Reviewer #1: Yes

Reviewer #2: Yes

5. Review Comments to the Author

Reviewer #1: I always have been supporting the idea of exploring the ischemic preconditioning as intervention that may cause a placebo effect. Most part of the studies about this topic show or try to show positive effects related to some vascular physiological event, while a strong motivational effect due to a simple manipulation of the subject should not be on of the main focus. In this sense, I agree with the idea of the study and my suggestions will be pointed bellow.

The manuscript is well written and the language is acceptable;

I would like to see a better link between title and short title; maybe change the short title

Modify abstract results: the use of IPC and SHAM at the same time hould be pointed, since the are many studies showing positive effects only with IPC

Change the conlcusion of the abstract and delete the reference to the literature; use your own data do conclusion;

Introduction

line 70-75: add a more detailed explanation about the rationale of why using IPC and SHAM together may explain the real IPC effects?

Methods

Sample

Add at least one performance parameter of the sample and not only the training history time;

In the explerimental design fig. Add more information and make the fig. fully self-explained, such as occlusion protocol and others.

In which leg the tests were carried out? Both? So, describe the dominant limb of subjects as well as gurantee this topic in the discussion;

I recommend one more paragraph about possible mechanisms underlying your experimental protocol and why we should or not the IPC or SHAM for enhacements in performance!!

Conclusion

Change to:

The adminstration of 3 cycles of 5 min cuff occlusion /reperfusion prior knee extension resistance exercise has no local effect and did not potentiate either the number of repetitions or MVIC in moderately RE trained men. A motivational negative effect seems to be related to this technique and, researchers and coaches should take this into account, before using IPC as a exercise enhacement tool, once a dissociation of IPC and placebo effect seems unlikely. Also , the inclusion of SHAM group in IPC studies is strongly recommended.

Reviewer #2: Title of the paper: “Does ischemic preconditioning really improve performance or it is just a placebo effect?” by de Souza HLR et al.

This study aimed at examining the effects of a simultaneous ischemic preconditioning (IPC) and SHAM intervention on the performance of knee extension resistance exercise. Authors found that applying IPC prior to a set of leg extension does not result in ergogenic effects. They concluded that the placebo effect seems to be the responsible for the ergogenic effect often reported for IPC in the literature.

General comments: the paper is in general of good quality. The text is clearly written, and the message easily gleaned by the reader.

Results dampen enthusiasm on the effects of ischemic preconditioning to improve exercise performance. In the recent past a plethora of studies have been provided to demonstrate the IPC is effective as an ergogenic aid. However, many of them lack from a rigorous scientific approach.

My main concern is that the protocol was limited to leg extension, so results are not applicable to real exercise in the field. I think that authors should highlight this limit.

Moreover, I suggest introducing the concept that the training status may play a role in the response to IPC. Athletes are probably less sensitive to this kind of maneuvers than sedentary or less fit subjects. For example, during field testing in trained runners it was not reported any ergogenic effect by IPC (refer for example to Tocco et al. Int J Sports Med 2015).

I offer some specific points:

Lines 66-69: not very clear this sentence. Please, reword it.

Lines 77-78: please, specify better what news you expected to find from your research.

lines 270-271: I would remove the sentence in brackets (i.e., alactic anaerobic capacity). The energetics of exercise is a quite complex matter, and this appears an oversimplification.

Lines 314-316: maybe some sentences explaining that also the training status

6. PLOS authors have the option to publish the peer review history of their article (what does this mean?). If published, this will include your full peer review and any attached files.

Reviewer #1: No

Reviewer #2: **Yes: **Antonio Crisafulli

---

## [Author Response · Author response to Decision Letter 0]

1 Apr 2021

Dear Editor and Reviewers: 

Thank you very much by your effort on our manuscript. Please receive our revised manuscript. We have followed the precious comments of the reviewers. 

Reviewer #1: I always have been supporting the idea of exploring the ischemic preconditioning as intervention that may cause a placebo effect. Most part of the studies about this topic show or try to show positive effects related to some vascular physiological event, while a strong motivational effect due to a simple manipulation of the subject should not be one of the main focus. In this sense, I agree with the idea of the study and my suggestions will be pointed bellow. 

The manuscript is well written and the language is acceptable;

R.: We appreciate all comments from the reviewer. We made all suggested changes and clarified the points that were not clear.

I would like to see a better link between title and short title; maybe change the short title.

R: We changed the short title as suggested.

Modify abstract results: the use of IPC and SHAM at the same time should be pointed, since the are many studies showing positive effects only with IPC.

R.: We include a brief sentence related to the intervention application that was performed at the same time.

Change the conclusion of the abstract and delete the reference to the literature; use your own data do conclusion

R.: The conclusion in the abstract was modified and the reference to the literature removed.

Introduction

line 70-75: add a more detailed explanation about the rationale of why using IPC and SHAM together may explain the real IPC effects? 

R.: We have added a more detailed explanation at this part of the text.

Methods

Sample

Add at least one performance parameter of the sample and not only the training history time;

R.: We added the mean value of the unilateral 1RM test of the subjects.

In the experimental design fig. Add more information and make the fig. fully self-explained, such as occlusion protocol and others. 

R.: We added more details in the experimental design figure.

In which leg the tests were carried out? Both? So, describe the dominant limb of subjects as well as guarantee this topic in the discussion;

R.: We added the percentage of right-handed in the subject’s section. We also included a sentence about limb dominance in the discussion section.

I recommend one more paragraph about possible mechanisms underlying your experimental protocol and why we should or not the IPC or SHAM for enhancements in performance!!

R.: We added one more paragraph according to the reviewer's suggestion.

Conclusion

Change to: The administration of 3 cycles of 5 min cuff occlusion /reperfusion prior knee extension resistance exercise has no local effect and did not potentiate either the number of repetitions or MVIC in moderately RE trained men. A motivational negative effect seems to be related to this technique and, researchers and coaches should take this into account, before using IPC as an exercise enhancement tool, once a dissociation of IPC and placebo effect seems unlikely. Also, the inclusion of SHAM group in IPC studies is strongly recommended.

R.: The text of conclusion was modified.

Reviewer #2: Title of the paper: “Does ischemic preconditioning really improve performance or it is just a placebo effect?” by de Souza HLR et al.

This study aimed at examining the effects of a simultaneous ischemic preconditioning (IPC) and SHAM intervention on the performance of knee extension resistance exercise. Authors found that applying IPC prior to a set of leg extension does not result in ergogenic effects. They concluded that the placebo effect seems to be the responsible for the ergogenic effect often reported for IPC in the literature.

General comments: the paper is in general of good quality. The text is clearly written, and the message easily gleaned by the reader.

Results dampen enthusiasm on the effects of ischemic preconditioning to improve exercise performance. In the recent past a plethora of studies have been provided to demonstrate the IPC is effective as an ergogenic aid. However, many of them lack from a rigorous scientific approach.

R.: We appreciate all comments from the reviewer. We reviewed all the points and also added all the suggestions made.

My main concern is that the protocol was limited to leg extension, so results are not applicable to real exercise in the field. I think that authors should highlight this limit.

R.: We added this limitation in the discussion section.

Moreover, I suggest introducing the concept that the training status may play a role in the response to IPC. Athletes are probably less sensitive to this kind of maneuvers than sedentary or less fit subjects. For example, during field testing in trained runners it was not reported any ergogenic effect by IPC (refer for example to Tocco et al. Int J Sports Med 2015).

R.: We added a brief comment about this topic on the discussion section.

I offer some specific points:

Lines 66-69: not very clear this sentence. Please, reword it.

R.: This point was clarified in the text.

Lines 77-78: please, specify better what news you expected to find from your research.

R.: We clarified this question in the previous paragraph.

lines 270-271: I would remove the sentence in brackets (i.e., alactic anaerobic capacity). The energetics of exercise is a quite complex matter, and this appears an oversimplification.

R.: Thanks. This part was deleted.

Lines 314-316: maybe some sentences explaining that also the training status

R.: This comment was addressed. We added an explanation about relation between training status and magnitude of placebo effect.

---

## [Decision Letter · Decision Letter 1]

12 Apr 2021

Does ischemic preconditioning really improve performance or it is just a placebo effect?

PONE-D-21-05373R1

Dear Dr. Marocolo,

We’re pleased to inform you that your manuscript has been judged scientifically suitable for publication and will be formally accepted for publication once it meets all outstanding technical requirements.

Kind regards,

Daniel Boullosa

Academic Editor

PLOS ONE

Additional Editor Comments (optional):

Reviewers' comments:

Reviewer's Responses to Questions

**Comments to the Author**

1. If the authors have adequately addressed your comments raised in a previous round of review and you feel that this manuscript is now acceptable for publication, you may indicate that here to bypass the “Comments to the Author” section, enter your conflict of interest statement in the “Confidential to Editor” section, and submit your "Accept" recommendation.

Reviewer #1: All comments have been addressed

Reviewer #2: All comments have been addressed

2. Is the manuscript technically sound, and do the data support the conclusions?

Reviewer #1: Yes

Reviewer #2: Yes

3. Has the statistical analysis been performed appropriately and rigorously? 

Reviewer #1: Yes

Reviewer #2: Yes

4. Have the authors made all data underlying the findings in their manuscript fully available?

Reviewer #1: Yes

Reviewer #2: Yes

5. Is the manuscript presented in an intelligible fashion and written in standard English?

Reviewer #1: Yes

Reviewer #2: Yes

6. Review Comments to the Author

Reviewer #1: (No Response)

Reviewer #2: I have no further questions.

Authors made a good job in revisiting their manuscript.

In my opinion the research deserves publication.

7. PLOS authors have the option to publish the peer review history of their article (what does this mean?). If published, this will include your full peer review and any attached files.

Reviewer #1: No

Reviewer #2: **Yes: **Antonio Crisafulli

---

## [Editor Report · Acceptance letter]

23 Apr 2021

PONE-D-21-05373R1 

Does ischemic preconditioning really improve performance or it is just a placebo effect? 

Dear Dr. Marocolo:

I'm pleased to inform you that your manuscript has been deemed suitable for publication in PLOS ONE. Congratulations! Your manuscript is now with our production department. 

Kind regards, 

on behalf of

Dr. Daniel Boullosa 

Academic Editor

PLOS ONE